# The Effects of Sarcopenia on Hip and Knee Replacement Surgery: A Systematic Review

**DOI:** 10.3390/medicina59030524

**Published:** 2023-03-07

**Authors:** Umile Giuseppe Longo, Sergio De Salvatore, Alessandro Borredon, Khazrai Yeganeh Manon, Anna Marchetti, Maria Grazia De Marinis, Vincenzo Denaro

**Affiliations:** 1Research Unit of Orthopaedic and Trauma Surgery, Fondazione Policlinico Universitario Campus Bio-Medico, Via Alvaro del Portillo, 00128 Rome, Italy; s.desalvatore@unicampus.it (S.D.S.); denaro@policlinicocampus.it (V.D.); 2Research Unit of Orthopaedic and Trauma Surgery, Department of Medicine and Surgery, Università Campus Bio-Medico di Roma, Via Alvaro del Portillo, 00128 Rome, Italy; 3Italy Department of Orthopaedic and Trauma Surgery, Campus Bio-Medico University, Via Alvaro del Portillo, 00128 Rome, Italy; 4Research Unit Nursing Science, Campus Bio-Medico di Roma University, Via Alvaro del Portillo, 00128 Rome, Italy; alessandroborredon@gmail.com (A.B.); a.marchetti@unicampus.it (A.M.); m.demarinis@unicampus.it (M.G.D.M.); 5Unit of Food Science and Nutrition, Department of Science and Technology for Humans and the Environment, Campus Bio-Medico University of Rome, 00128 Rome, Italy; m.khazrai@policlinicocampus.it

**Keywords:** functional outcomes, knee, hip, joint replacement, sarcopenia

## Abstract

Sarcopenia is a progressive and generalized skeletal muscle disorder associated with poor outcomes and complications, including falls, fractures, physical disability, and death. The aim of this review is to assess the possible influence of sarcopenia on outcomes of sarcopenia in patients who underwent knee or hip replacement. A systematic review was performed using the Preferred Reporting Items for Systematic Reviews and Meta-analyses (PRISMA) guidelines. Medline, EMBASE, Scopus, CINAHL, and CENTRAL bibliographic databases were searched. General study characteristics extracted were: primary author and country, year of publication, type of study, level of evidence (LOE), sample size, mean age, gender, follow-up, type of surgery, diagnosis, and outcomes. At the final screening, five articles met the selection criteria and were included in the review. Sarcopenia influences the Barthel Index (BI), which is significantly lower compared to patients without sarcopenia, which indicates that the patient is subjected to a worsening of this condition that can influence their normal life since they will become dependent on someone else. No difference in mortality rate was found was found between the studies. This systematic review addressed the possible role of sarcopenia in patients undergoing joint replacement surgery. Despite the lack of high-quality literature on this topic, a general trend in considering sarcopenia as a negative factor for quality of life in joint replacement patients was reported. However, the lack of significant results means it is not possible to report useful conclusions.

## 1. Introduction

Sarcopenia is a progressive and generalized skeletal muscle disorder that has been linked to various poor outcomes and complications, including falls, fractures, physical disability, and even death [1,2,3,4]. It is a common condition in both sexes, with prevalence rates ranging from 9% to 18% in individuals between 65 and 70 years old [5]. To diagnose sarcopenia, two of three criteria are required: low muscle strength, low muscle quantity and quality, and low physical performance [1,6]. Several factors have been implicated in the pathophysiology of sarcopenia, including decreased caloric intake, inactivity, hormonal decline, loss of anabolic stimuli, insulin resistance, muscle fiber denervation, elevated inflammatory cytokine levels (IL-6 or TNF-alpha), and increased myostatin levels [1,2,7]. Decreased caloric intake can lead to a lack of nutrients necessary for muscle maintenance and growth. Inactivity, especially in older individuals, can result in muscle atrophy and weakness [1,2,7]. Hormonal decline, such as a decrease in testosterone or growth hormone levels, can also contribute to muscle loss. The loss of anabolic stimuli, such as exercise or adequate protein intake, can further exacerbate muscle wasting. Insulin resistance, a condition commonly seen in individuals with type 2 diabetes, can also contribute to muscle loss [1,2,7]. Muscle fiber denervation, which can occur with age or neurological diseases, can lead to a reduction in muscle mass and strength. Elevated levels of inflammatory cytokines, such as IL-6 or TNF-alpha, have been associated with muscle wasting and increased risk of sarcopenia. Finally, increased myostatin levels, a protein that inhibits muscle growth, have been implicated in the development of sarcopenia. These factors can act in combination or individually to contribute to the development and progression of sarcopenia [1,2,7]. Patients with sarcopenia may experience a loss of up to 15% of their total muscle mass, and those who undergo orthopedic surgical procedures may experience muscle mass loss of up to 44% [6,7]. Hospitalization, recovery from illness, and periods of physical inactivity can exacerbate this muscle loss.

Bioelectrical impedance analysis (BIA) is an accessible method to diagnose sarcopenia as it is relatively affordable and highly portable, although it can be affected by the degree of hydration and edemas [8,9]. Ultrasound (US) is another accessible method that allows for the assessment of muscle mass and quality, but it cannot measure muscle mass directly [10]. Dual-energy X-ray absorptiometry (DEXA) provides a more precise approach to diagnosing sarcopenia by measuring the attenuation of two different types of X-rays in the body [9]. DEXA can differentiate three types of bodily composition (fat, bone mineral, and lean tissue) and provides an accurate measurement of muscle mass. However, it is a more expensive method that requires technical knowledge and exposes patients to radiation [11].

As the population continues to age, there has been a notable increase in the average age of patients requiring joint replacement surgery [12,13,14]. Joint replacement surgery has been shown to improve the quality of life of those suffering from joint disorders in numerous studies [15]. Understanding the potential impact of sarcopenia on joint replacement surgery outcomes is crucial for optimizing patient outcomes and enhancing the quality of life for those undergoing the procedure. As sarcopenia often accompanies osteoarthritis, an increased incidence of elderly patients with sarcopenia undergoing joint replacement surgery is expected in the coming years [16,17].

To our knowledge, no studies have investigated the role of sarcopenia in patients who underwent joint replacement surgery. Therefore, the aim of this review is to assess the possible influence on outcomes of sarcopenia in patients who underwent knee or hip replacement.

## 2. Materials and Methods

The present paper focused on studies concerning sarcopenia in patients undergoing joint replacement surgery. The Preferred Reporting Items for Systematic reviews and Meta-Analyses (PRISMA) guidelines were used to improve the reporting of the review.

### 2.1. Study Selection

The research question was formulated using a PICOS approach: Patient (P); Intervention (I); Outcome (O); and Study design (S). This study selected those articles that described patients with sarcopenia (P), that had undergone knee or hip replacement surgery (I). The aim was to value how the sarcopenia can influence the outcomes in joint replacement population (O). The presence of sarcopenia was assessed by body mass index (BMI) and handgrip strength (HGS), Harris hip score (HHS), appendicular skeletal muscle mass (ASM), total body fat, low muscle mass (identified by a cutoff value of AMI < 6.12 kg/m^2^), skeletal muscle mass index (SMMI), gluteal muscle power, lean body mass. For this purpose (S), randomized studies (RCT) and non-randomized controlled studies (NRCT) such as prospective (PS), retrospective (RS), cross-sectional (CS), observational studies (OS), case series (CS), and case–control (CC) studies were included.

### 2.2. Inclusion Criteria

Only articles published in English were screened. Peer-reviewed articles of each level of evidence according to Oxford classification were considered. Studies reporting patients over 60 years undergoing a joint replacement (knee and hip) were included. Patients with a certified diagnosis of sarcopenia assessed by two of three of the following criteria: low muscle strength, low muscle quantity or quality, low physical performance were considered eligible for the study.

### 2.3. Exclusion Criteria

Reviews, books, protocol studies, case reports, technical notes, letters to editors, instructional courses, in vitro, and cadaver studies were excluded. In addition, articles reporting outcomes of patients with normal muscular control, certain joint infection were excluded. Moreover, studies in which patients refused a dual-energy X-ray absorptiometry (DXA) study, use of medication interacting with muscle metabolism and mobility of the limbs, such as chronic obstructive pulmonary disease, peripheral arterial disease, severe cardiovascular impairment, neurologic disorders, coagulation diseases, and malignant disease, uncontrolled hypertension, any cardiovascular or pulmonary disease that would prevent them from engaging in an exercise study, or neurological or cognitive impairment, if they received hip surgery as a result of a condition other than a primary hip fracture, including osteoarthritis, trauma, tumor, infection, and avascular necrosis of the femoral heads, who failed to complete the sarcopenia assessment using dual-energy DXA, evidence of malignancy by preoperative CT, previous stroke, and unable to cooperate (dementia, delirium, depression, or other conditions) were excluded.

### 2.4. Search

A systematic review was performed using the Preferred Reporting Items for Systematic Reviews and Meta-analyses (PRISMA) guidelines. Medline, EMBASE, Scopus, CINAHL, and CENTRAL bibliographic databases were searched using the following keywords (isolated or combined): “Muscle”, “Weight”, “Strength”, “Sarcopenia”, “Joint”, “Knee”, “hip”, “Arthroplasty”, “Replacement”, “Prostheses”, “dual-energy X-ray absorptiometry (DXA)”, and “Bioelectrical Impedance Analysis (BIA)”. More studies were searched among the reference lists of the selected papers. The search was performed by two of the authors (A.B. and S.D.S.) from January to February 2022 and articles from the inception of the database to March 2022 were searched.

### 2.5. Data Collection Process

Data extraction was performed by two independent reviewers (A.B. and S.D.S.) and differences were reconciled by mutual agreement. In cases of disagreement on inclusion or exclusion of articles, a third reviewer (UGL) was consulted. The same authors (A.B. and S.D.S.) performed the review and organization of the titles in order to limit the bias.

The reviewers used the following screening approach: title and abstract were reviewed first, then the full articles. The full text of papers not excluded was evaluated and eventually selected after a discussion between the reviewers. The number of articles included or excluded are registered and reported in the PRISMA flowchart (Figure 1).

### 2.6. Data Items

General study characteristics extracted were: primary author and country, year of publication, type of study, level of evidence (LOE), sample size, mean age, sex, follow-up, type of surgery, diagnosis, outcomes. All the results are summarized in Table 1 and Table 2.

The statistical analysis was performed by one of the author (S.D.S.).

### 2.7. Study Risk of Bias Assessment

Two authors (A.B. and S.D.S.) independently assessed the potential assessed risk of bias of the studies included using the MINORS, a methodological index for non-randomized studies. The items were scored 0 if not reported; 1 when reported but inadequate; and 2 when reported and adequate. Low risk of bias was considered when studies fulfilled all MINORS criteria; conversely, high risk of bias was considered in all other studies. Consensus was reached by the two reviewers (A.B. and S.D.S.) when there was difference in opinion on an item. If no consensus was reached, the independent opinion of a third reviewer was decisive (U.G.L.).

## 3. Results

### 3.1. Study Selection

The literature search identified 396 articles. No additional studies were found in the grey literature and no unpublished studies were retrieved. Duplicate removal resulted in the exclusion of 36 studies, leaving 366 articles for screening. In total, 340 articles were excluded based on the title and abstract. A further 26 articles were screened by full text and 21 were excluded. At the final screening, five articles met the selection criteria and were included in the review. The PRISMA flowchart of the literature search is reported in Figure 1.

### 3.2. Study Characteristics

The included articles were one RCT [19] and four NRCTs (2 CC, 1 PS, and 1 OS). Studies were published between 2018 [8] and 2020 [2,6,18,19]. The mean age ranged between 60 [18,19] and 80 years [6]. Follow-up of the included studies ranged from 6 days [7] to 106 months [2]. Hip replacement was the most common procedure performed [12,18,19]. The most common scores adopted were BMI [1,2,6,7,12,18,19], HGS [6,18], ASM [6,18,19], total body fat [7,18], and SMMI [6,7]. Other scores used were low muscle mass, HHS, lean body, and gluteal muscle power [2,7,12,18,19]. A summary of the characteristics of the included studies is reported in Table 1.

### 3.3. Quality Assessment

The RCT included [19] reported “some concerns risk of bias” using the ROB-2 tool. The MINORS tool was adopted to assess the Quality of Evidence of the included NRCT papers. All of these studies had a low risk of bias [1,2,6,7,12,18,19,20]. The MINORS score is reported in Table 2. The risk of bias assessments for the RCT is reported in Figure 2.

### 3.4. Outcome: PROMS Improvement

Chen et al. [16] reported a comparison between patients with and without sarcopenia in the preoperative period without a follow-up, suggesting a notable difference between groups, as evidenced in Table 3.

Sarcopenic patients reported an average value of the Barthel Index (BI) of 80.65 ± 25.26, while the patients without sarcopenia presented a value of 90.29 ± 19.60 resulting in almost total independence.

Bae et al. [2] reported the BI differences between a normal patient and osteosarcopenic patient, suggesting a relation between sarcopenia and osteoporosis. Additionally, the follow-up periods were reported, which allows for assessing over time the change in BI. Therefore, it seems to be the case that sarcopenia influences the BI, which is significantly lower compared to normal patients, which indicates that the patient is subjected to a worsening of this condition that can influence their normal life since they will become dependent on someone else.

### 3.5. Outcome Mortality Rate

Mortality in the sarcopenia group appears to be higher as reported by Yoo et al. [20] (mortality: 15.1% sarcopenia, 10.3% controls). This is in contrast with Kim et al. [21] and Bae et al. [2] who reported no appreciable difference in mortality in the 1-year follow-up period. It is pivotal to note that Kim et al. [21] reported a spike in the mortality rate in a 5-year follow-up (82.7%), while Chen et al. [19] reported a 5% mortality rate in a 6-month follow-up time.

## 4. Discussion

This study aimed to conduct a systematic review on the potential influence of sarcopenia on joint replacement surgery outcomes. Sarcopenia has a significant impact on patient-reported outcome measures (PROMs) and may increase the mortality rate. However, the association between sarcopenia and mortality rate requires further investigation. The findings of this study underscore the importance of early detection and management of sarcopenia in older adults to improve their quality of life and reduce the risk of adverse outcomes.

The included studies showed a higher prevalence of sarcopenia in men compared to women [1,2]. The higher rate of sarcopenia in men is likely due to the age of presentation, type of injury, and poorer general health condition. However, this is in contrast with two other studies that reported a higher incidence of sarcopenia in women [6,18]. Nevertheless, the real association between sex and sarcopenia has not been fully elucidated [6]. Chen et al. reported that male sex, lower BMI, and lower handgrip strength can be reliable predictors and could be used to screen older patients at risk of sarcopenia [18,21]. Furthermore, sarcopenia is associated with other comorbidities, particularly osteoporosis [13,22,23]. Bae et al. reported a group of patients with both pathologies [2]. Sarcopenia is an independent predictor of poor functional recovery and a decrease in life quality expectancy at 6 months after surgery. However, Chen and colleagues [16] did not provide significant results.

Kouw et al. reported a high rate of muscle disuse due to prolonged hospitalizations, which can exacerbate sarcopenia [7]. Surgery, associated physical or mental stress, and reduced food intake during hospitalization are likely to further aggravate skeletal muscle loss in clinical practice [24,25]. Therefore, long-term hospitalization might lead to significant loss in skeletal muscle mass and strength in older patients, posing a major threat to fully regaining physical function after discharge. Screening patients for sarcopenia before surgery could aid in the proper management of these patients using minimally invasive surgery and fast-track discharge protocols, reducing the risk of a prolonged hospital stay [6].

The findings regarding PROMs improvement indicated that patients with sarcopenia had a significantly lower Barthel Index (BI) compared to normal patients, suggesting that sarcopenia has an adverse effect on patients’ quality of life and may lead to dependency on others. The results were consistent across the studies, suggesting a robust association between sarcopenia and BI. According to Bae and colleagues [2], as well as Chen, there was a reduction in BI and quality of life at 6 months after surgery [18]. Therefore, both authors suggest that sarcopenia is associated with lower outcomes and quality of life in patients who underwent joint replacement.

Regarding the mortality rate, one study reported a higher mortality rate in the sarcopenia group compared to controls, while two other studies reported no appreciable difference in mortality rates between the two groups. However, one of the studies that reported no difference in mortality rate had a relatively short follow-up period, while another study reported a spike in mortality rate in a 5-year follow-up. Kim et al. and Yoo et al. reported a significant difference between the postoperative 5-year mortality rates of sarcopenic and healthy patients, with a 30% higher mortality rate in the former group [26,27]. These data require further prolonged studies to assess the risk of death of sarcopenic patients who underwent joint replacement surgery.

### Limitations

The limitations of this review are mainly based on the low number of studies included. Furthermore, the level of evidence of the included studies was low; therefore, there is a lack of high-quality literature on this topic. Moreover, the population was not homogeneous. The studies included in the review used different definitions of sarcopenia, different outcome measures, and different follow-up periods, which makes it difficult to compare the results across studies. Furthermore, the review only included studies that were published in English and indexed in the selected databases. Studies published in other languages or in non-indexed databases may have been missed. The studies included in the review did not report information on confounding variables, such as comorbidities or medication use, which may have influenced the outcomes. Moreover, the study by Kouw and colleagues reported conclusions with just 6 days of follow-up; therefore, the results could not be considered relevant [7]. Furthermore, there is only one study on total knee replacement, and regarding the four studies on hip replacement, three of them concerned hip replacement after hip fracture.

In the process of searching for these articles, it was identified as a limiting factor that most of the literature available report only on the change in sarcopenia post-surgery without reporting the proper change in sarcopenia itself.

Specific cutoff values for HGS differ between the Asian working group (AWGS) for sarcopenia and the revised European working group on sarcopenia in older people (EWGSOP2) criteria. This makes the comparison between work performed in Asian populations and other countries difficult. This variation may be explained in part by inherent differences in muscle strength and muscle quantity among people of different ethnicities [6].

Lastly, there is a lack of significant results. While the review reports a general trend in considering sarcopenia as a negative factor for quality of life in joint replacement patients, the lack of significant results makes it difficult to draw useful conclusions.

## 5. Conclusions

This review examined the potential impact of sarcopenia in joint replacement patients. Although there is limited high-quality research available, several studies suggest that sarcopenia may negatively affect patients’ quality of life. This result agrees with other studies reported. However, the lack of significant results prevents drawing definitive conclusions. Given the increasing number of joint replacement surgeries due to aging populations, further high-quality studies are needed to investigate sarcopenia’s impact. Additionally, developing standardized diagnostic criteria is essential for producing consistent research and facilitating cross-country comparisons.

## Figures and Tables

**Figure 1 medicina-59-00524-f001:**
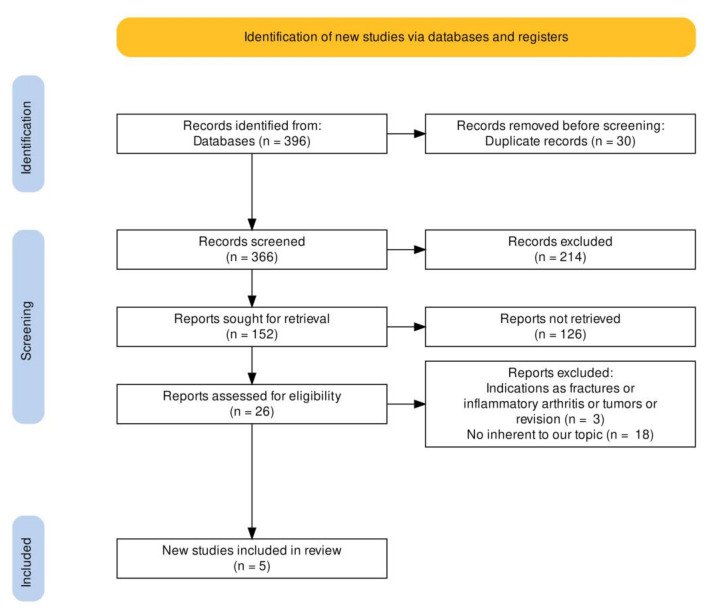
PRISMA flowchart.

**Figure 2 medicina-59-00524-f002:**
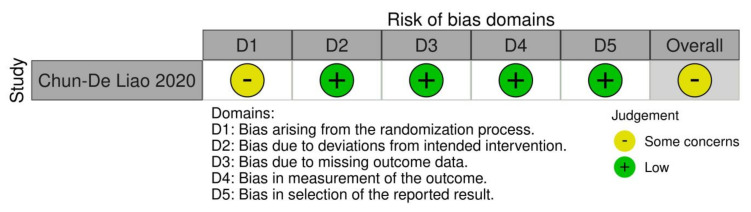
ROB2 assessment of the included study [19].

**Table 1 medicina-59-00524-t001:** Characteristics of the included studies.

Author, Year of Publication, and Country	Type of Study and Level of Evidence (LOE)	Mean Age	Sample Size and Sex (Female)	Follow-Up
Bae 2020, Republic of Korea [2]	Case–control study, III	Group 1: 82.8 ± 6.6 Group 2: 83.3 ± 6.0	109 (87): Group 1: 64 with sarcopenia Group 2: 45 normal group	108 m
Chen 2020, Taiwan [18]	Prospective study, III	Group 1: 83.49 ± 9.77 Group 2: 77.99 ± 8.80	139 (103) Group 1: 69 with sarcopenia Group 2: 70 without sarcopenia	12 m
Chun-De Liao 2020, Taiwan [19]	RCT, I	Group 1 (experimental): 72.22 ± 7.75 Group 2 (control): 69.79 ± 6.72	40 (not specified) group 1: 20 group 2: 20	4 m
Kouw2018, The Netherlands [7]	Observational study, III	74.7 ± 0.8	26 (19)	6 d
Laubscher 2020, South Africa [6]	Case–control study, II	Group 1: 78 ± 10 Group 2: 71 ± 8	65 (39) group 1: 34 (24) with sarcopenia group 2: 31 (15) without sarcopenia	4 m

**Table 2 medicina-59-00524-t002:** MINORS score of included studies.

Author	Clearly Stated Aim	Inclusion of Consecutive Patients	Prospective Data Collection	Endpoints Appropriate to Study Aim	Unbiased Assessment of Study Endpoint	Follow-Up Period Appropriate to Study Aim	<5% Lost to Follow-Up	Prospective Calculation of Study Size	Adequate Control Group	Contemporary Groups	Baseline Equivalence of Groups	Adequate Statistical Analyses	Total Score (…/24)
Bae 2020	2	0	2	2	2	2	2	2	2	2	2	2	22
Chen 2020	2	0	0	2	NA	2	2	2	0	0	0	2	12
Kouw 2018	2	2	0	2	2	0	2	0	0	0	0	2	12
Laubscher 2020	2	0	2	2	2	1	2	2	0	0	0	2	15

**Table 3 medicina-59-00524-t003:** Quantitative results of included studies.

Author and Year	Type of Surgery	Diagnosis	Outcome Measures	Mean Changes in Outcome Variables		
				Sarcopenia Group	Non-Sarcopenia Group	*p*-Value
Bae 2020, [2]	Hip replacement	Hip fracture	BMI; HHS; BI;	BMI 23.9 ± 8.0 HHS: Preoperative 78.2 ± 5.1 6 weeks 56.2 ± 7.5 3 months 59.1 ± 5.9 1 year 64.7 ± 8.5 BI: Preoperative 73.9 ± 6.8 6w 52.4 ± 8.8 3 m 58.0 ± 7.6 1 y 62.9 ± 6.4	BMI 23.9 ± 3.1 HHS: Preoperative 81.4 ± 6.0 6 w 66.8 ± 7.1 3 m 70.2 ± 6.6 1 y 74.0 ± 7.3 BI: Preoperative 76.0 ± 8.1 6 w 61.4 ± 7.8 3 m 68.3 ± 7.1 1 y 72.8 ± 7.3	*p* = 0.286 * *p* = 0.092 *p* < 0.001 * *p* < 0.001 * *p* < 0.001 * *p* = 0.167 *p* < 0.001 * *p* < 0.001 * *p* < 0.001 *
Chen 2020, [18]	Hip replacement	Hip fracture	BMI; HGS(Kg); BI; Total body fat (%) EQ-5D	Preoperative BMI: 20.94 ± 3.27 Preoperative HGS: 9.84 ± 5.4 Preoperative BI: 80.65 ± 25.26 Preoperative Total body fat: 31.52 ± 8.43 Preoperative EQ-5D: 0.78 ± 0.21	Preoperative BMI: 24.34 ± 3.09 Preoperative HGS: 13.84 ± 9.27 Preoperative BI: 90.29 ± 19.60 Preoperative Total body fat: 36.52 ± 6.36 Preoperative EQ-5D: 0.88 ± 0.18	*p* < 0.001 * *p* < 0.001 * *p* < 0.001 * *p* < 0.001 * *p* < 0.001 *
Chun-De Liao 2020 [19]	Total knee replacement	Knee osteoarthritis	BMI; AMI; WOMAC PAIN WOMAC PF	Group 1: BMI t0: 28.27 ± 3.25 AMI: 6.22 ± 1.10 WOMAC PAIN: t0:12.5 ± 3.23 t2–t0: −6.95 ± 2.55 WOMAC PF: t0: 4.50 ± 1.82 t2–t0: −0.95 ± 1.76	Group 2: BMI t0: 27.60 ± 3.64 AMI: 5.95 ± 0.99 WOMAC PAIN: t0:10.28 ± 3.32 t2–t0: −5.55 ± 1.23 WOMAC PF: t0: 3.45 ± 2.11 t2–t0: 0.60 ± 2.09	*p*-Value not reported
Kouw 2018, [7]	Hip replacement	Hip osteoarthritis	BMI; Lean body mass, kg (%); Fat mass, %; SMMISF-36	BMI: 28.0 ± 0.8 lean body mass: 47.6 ± 1.9; fat mass: 34.0 ± 1.2; SMMI: 7.6 ± 0.3; SF36: not reported	SF36: not reported	*p*-Value not reported
Laubscher 2020, [6]	Hip replacement	Hip fractures	BMI; HGS; ASM; SMI. SURGICAL OUTCOMES NOT REPORTED	Group 1: BMI: 21 ± 4 Low HGS **: 34 Normal HGS: 0 Low ASM ***: 34 Normal ASM: 0 Low SMI ****: 28 Normal SMI: 6	Group 2: BMI: 24 ± 7 Low HGS: 34 Normal HGS: 0 Low ASM: 34 Normal ASM: 0 Low SMI: 28 Normal SMI: 6	*p* = 0.013 *

HGS: handgrip strength; BMI: body mass index; ASM: appendicular skeletal muscle mass; SMMI: skeletal muscle mass index; AMI: appendicular mass index; HHS: Harris hip score; BI Barthel Index; WOMAC PF: Western Ontario and McMaster Universities Osteoarthritis Index—Physical Function. * Low muscle mass was identified by a cutoff value of AMI <6.12 kg/m^2^. ** Low HGS refers to handgrip strength <16 kg in women and <27 kg in men. *** Low ASM refers to an ASM of <15 kg in women and <20 kg in men. **** Low SMI refers to <5.5 kg/m^2^ for women and <7 kg/m^2^, for men.

## Data Availability

The data presented in this study are available on request from the corresponding author.

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
