# Peer review of "The Effects of Sarcopenia on Hip and Knee Replacement Surgery: A Systematic Review"

_medicina, 2023, doi:10.3390/medicina59030524_

Round 1
Reviewer 1 Report
Review
Many thanks to the authors for having presented a so interesting systematic review about
"The Effects of Sarcopenia on Joint Replacement Surgery: A Systematic Review.
Before resubmitting the revision version of the article, please read the editorial rules carefully, and check other editorial aspects, such as: text alignment (lacking), text justification at the head (lacking), etc. The language is good that the manuscript does not need to be corrected by a person of English mother tongue.
Abstract
The abstract is well structured, and it contains the main information of the study.
Key words
Please provide them in alphabetic order.
Background
The introduction is well structured, containing the main aims of the study. However, a few lines about the improvement of life quality among general population after Joint Replacement Surgery should be add to better introduce the topic, quoting also:
· Quality of life outcomes in patients undergoing knee replacement surgery: longitudinal findings from the QPro-Gin study. BMC Musculoskelet Disord. 2020 Jul 4;21(1):436. doi: 10.1186/s12891-020-03456-2.
Methods
This section contains enough information to understand and possibly repeat the study.
Results
The results presented are complete. However, some tables should be integrate to better represent the results of the study.
Statistical analysis
The statistical analysis is accurate and detailed. Please provide who performed the analysis: an independent statistician or the same authors?
Discussion
The discussion is detailed and explanatory. The length and content of the discussion communicates the main information of the paper. The discussion explains the results relative to prior publications, but it does not recognize sufficient limitations of the manuscript. Please, improve this section.
Lines 179-181: “Therefore, short-term hospitalization might be accompanied by a significant loss in skeletal muscle mass and strength in older patients, posing a major threat to fully regaining physical function after discharge. Screening sarcopenia patients before surgery, could aid to proper manage these patients using minimally invasive surgery and fast track discharge protocols, reducing the risk of prolonged hospital stay.” I guess there is an error, please correct the underlined words: long-term?
Conclusions
The conclusions reflect and refer to the results of the study.
References
The references are up to date. However, please delete those before 2010 if not strictly essential, eventually replacing them with newer ones and integrate them as suggested previously.
Tables
The number and quality of tables should be improved to transmit the main information of the paper in relation to results sections.
Author Response
Dear reviewer,
We would like to thank you for the helpful comments and suggestions. We have revised the paper accordingly and hope that the work is now ready for publication. The changes are itemized below with our comments to the reviewer’s suggestions. Changes made in the text are highlighted in yellow in the original manuscript.
Reviewer 1
Review
Many thanks to the authors for having presented a so interesting systematic review about
"The Effects of Sarcopenia on Joint Replacement Surgery: A Systematic Review.
Before resubmitting the revision version of the article, please read the editorial rules carefully, and check other editorial aspects, such as: text alignment (lacking), text justification at the head (lacking), etc. The language is good that the manuscript does not need to be corrected by a person of English mother tongue.
Thanks for the comment. We are honored you appreciated our paper. We uploaded it as free format as reported by the guidelines of the system. However we agree to you and we formatted the. Manuscript with editorial rules about text alignment.
Abstract
The abstract is well structured, and it contains the main information of the study.
Thanks for the comment
Key words
Please provide them in alphabetic order.
Thanks for the comment. We revised the manuscript accordingly.
Background
The introduction is well structured, containing the main aims of the study. However, a few lines about the improvement of life quality among general population after Joint Replacement Surgery should be add to better introduce the topic, quoting also:
- Quality of life outcomes in patients undergoing knee replacement surgery: longitudinal findings from the QPro-Gin study. BMC Musculoskelet Disord. 2020 Jul 4;21(1):436. doi: 10.1186/s12891-020-03456-2.
Thanks for the comment. We revised the manuscript accordingly. The citation you provided improved the overall quality of our paper.
Methods
This section contains enough information to understand and possibly repeat the study.
Thanks for the comment.
Results
The results presented are complete. However, some tables should be integrate to better represent the results of the study.
Thanks for the comment. We tried to layout better the tables to be more understandable for the readers
Statistical analysis
The statistical analysis is accurate and detailed. Please provide who performed the analysis: an independent statistician or the same authors?
Thanks for the comment. The statistical analysis was performed by one of the author SDS. We improved the manuscript accordingly
Discussion
The discussion is detailed and explanatory. The length and content of the discussion communicates the main information of the paper. The discussion explains the results relative to prior publications, but it does not recognize sufficient limitations of the manuscript. Please, improve this section.
Lines 179-181: “Therefore, short-term hospitalization might be accompanied by a significant loss in skeletal muscle mass and strength in older patients, posing a major threat to fully regaining physical function after discharge. Screening sarcopenia patients before surgery, could aid to proper manage these patients using minimally invasive surgery and fast track discharge protocols, reducing the risk of prolonged hospital stay.” I guess there is an error, please correct the underlined words: long-term?
Thanks for the comment. We revised the manuscript accordingly.
Conclusions
The conclusions reflect and refer to the results of the study.
Thanks for the comment.
References
The references are up to date. However, please delete those before 2010 if not strictly essential, eventually replacing them with newer ones and integrate them as suggested previously.
Thanks for the comment. We revised the manuscript accordingly.
Tables
The number and quality of tables should be improved to transmit the main information of the paper in relation to results sections.
Thanks for the comment. We tried to layout better the tables to be more understandable for the readers
Reviewer 2 Report
The paper has severe English mistakes that make it difficult to comprehend. There are numerous grammatical errors, poor sentence construction, and incorrect use of vocabulary that make the paper difficult to follow. Additionally, the paper lacks a clear and concise conclusion section that summarizes the findings of the systematic review.
Furthermore, the introduction and discussion sections are far too short for a systematic review. The introduction should provide a thorough background of the topic, while the discussion section should offer a comprehensive analysis of the results obtained. However, the paper only briefly touches on these areas, leaving the reader without a clear understanding of the scope of the topic and the implications of the findings.
The paper's methodology is correct, and the systematic review is well-conducted. The authors has used the PRISMA guidelines to perform a comprehensive literature search, ensuring that the search results are unbiased and relevant.
Overall, "The Effects of Sarcopenia on Joint Replacement Surgery: A Systematic Review" is a promising paper that explores a topic of great interest. The paper's language mistakes, lack of a conclusion, and brief introduction and discussion sections detract from its overall impact. With further revisions and edits, the paper could become a valuable contribution to the field of joint replacement surgery and sarcopenia. Below are my suggestions:
Title
It is not all joint replacement surgery. Please revise the title with hip and knee only.
Abstract
In line 17, "influence on outcomes" should be "influence of sarcopenia on outcomes".
In line 21, "sex" should be "gender".
In line 23, "It seems to be the case that" is unnecessary and should be removed.
In line 24, "normal patients" should be "patients without sarcopenia".
In line 25, "No according about mortality rate" is unclear and should be rephrased to "No difference in mortality rate was found".
In line 27, "the possible role sarcopenia" should be "the possible role of sarcopenia".
In line 28, "makes not possible to report" should be "makes it not possible to report".
The text could benefit from clearer transitions between sentences and paragraphs to improve readability.
Introduction
In sentence 35, "people" should be replaced with "individuals".
In sentence 36, "low muscle quantity or quality" should be "low muscle quantity and quality".
In sentence 43, "disorder" should be "distorted".
In sentence 47, "X Ray" should be "X-ray" or "radiographs".
In sentence 49, "has is expansive" should be "is expensive".
Methods
Sentence 57, "patients" and not "patient"
Otherwise, very good
Results are ok
Discussion section must be improved with more relevant data.
The conclusions are missing. Please upload a version that has all the required sections of a scientifically correct written manuscript.
References are up to date and relevant
Author Response
Dear reviewers,
We would like to thank you for the helpful comments and suggestions. We have revised the paper accordingly and hope that the work is now ready for publication. The changes are itemized below with our comments to the reviewer’s suggestions. Changes made in the text are highlighted in yellow in the original manuscript.
Reviewer 2
The paper has severe English mistakes that make it difficult to comprehend. There are numerous grammatical errors, poor sentence construction, and incorrect use of vocabulary that make the paper difficult to follow. Additionally, the paper lacks a clear and concise conclusion section that summarizes the findings of the systematic review.
Thanks for the comment. The paper has been revised by a native English speaker. We improved the conclusion section accordingly.
Furthermore, the introduction and discussion sections are far too short for a systematic review. The introduction should provide a thorough background of the topic, while the discussion section should offer a comprehensive analysis of the results obtained. However, the paper only briefly touches on these areas, leaving the reader without a clear understanding of the scope of the topic and the implications of the findings.
Thanks for the comment. We improved the introduction and the discussion accordingly.
The paper's methodology is correct, and the systematic review is well-conducted. The authors has used the PRISMA guidelines to perform a comprehensive literature search, ensuring that the search results are unbiased and relevant.
Thanks for the comment.
Overall, "The Effects of Sarcopenia on Joint Replacement Surgery: A Systematic Review" is a promising paper that explores a topic of great interest. The paper's language mistakes, lack of a conclusion, and brief introduction and discussion sections detract from its overall impact. With further revisions and edits, the paper could become a valuable contribution to the field of joint replacement surgery and sarcopenia. Below are my suggestions:
Thanks for the comment. We improved the overall quality of our paper thanks to your suggestions.
Title
It is not all joint replacement surgery. Please revise the title with hip and knee only.
Thanks for the comment. We revised the manuscript accordingly.
Abstract
In line 17, "influence on outcomes" should be "influence of sarcopenia on outcomes".
In line 21, "sex" should be "gender".
In line 23, "It seems to be the case that" is unnecessary and should be removed.
In line 24, "normal patients" should be "patients without sarcopenia".
In line 25, "No according about mortality rate" is unclear and should be rephrased to "No difference in mortality rate was found".
In line 27, "the possible role sarcopenia" should be "the possible role of sarcopenia".
In line 28, "makes not possible to report" should be "makes it not possible to report".
The text could benefit from clearer transitions between sentences and paragraphs to improve readability.
Thanks for the comment. We revised the manuscript accordingly.
Introduction
In sentence 35, "people" should be replaced with "individuals".
In sentence 36, "low muscle quantity or quality" should be "low muscle quantity and quality".
In sentence 43, "disorder" should be "distorted".
In sentence 47, "X Ray" should be "X-ray" or "radiographs".
In sentence 49, "has is expansive" should be "is expensive".
Thanks for the comment. We revised the manuscript accordingly.
Methods
Sentence 57, "patients" and not "patient"
Otherwise, very good
Thanks for the comment. We revised the manuscript accordingly.
Results are ok
Thanks for the comment.
Discussion section must be improved with more relevant data.
Thanks for the comment. We revised the manuscript accordingly.
The conclusions are missing. Please upload a version that has all the required sections of a scientifically correct written manuscript.
Thanks for the comment. We revised the manuscript accordingly.
References are up to date and relevant
Thanks for the comment
Round 2
Reviewer 2 Report
Authors have now made the requested changes and the article is ready to be published.